# ON THE REPRODUCIBILITY OF NEURAL NETWORK PREDICTIONS

## ABSTRACT

Standard training techniques for neural networks involve multiple sources of randomness, e.g., initialization, mini-batch ordering and in some cases data augmentation. Given that neural networks are heavily over-parameterized in practice, such randomness can cause *churn* – disagreements between predictions of the two models independently trained by the same algorithm, contributing to the 'reproducibility challenges' in modern machine learning. In this paper, we study this problem of churn, identify factors that cause it, and propose two simple means of mitigating it. We first demonstrate that churn is indeed an issue, even for standard image classification tasks (CIFAR and ImageNet), and study the role of the different sources of training randomness that cause churn. By analyzing the relationship between churn and prediction confidences, we pursue an approach with two components for churn reduction. First, we propose using *minimum entropy regularizers* to increase prediction confidences. Second, we present a novel variant of co-distillation approach (Anil et al., 2018) to increase model agreement and reduce churn. We present empirical results showing the effectiveness of both techniques in reducing churn while improving the accuracy of the underlying model.

## 1 INTRODUCTION

Deep neural networks (DNNs) have seen remarkable success in a range of complex tasks, and significant effort has been spent on further improving their predictive *accuracy*. However, an equally important desideratum of any machine learning system is *stability* or *reproducibility* in its predictions. In practice, machine learning models are continuously (re)-trained as new data arrives, or to incorporate architectural and algorithmic changes. A model that changes its predictions on a significant fraction of examples after each update is undesirable, even if each model instantiation attains high accuracy.

Reproducibility of predictions is a challenge even if the architecture and training data are fixed across different training runs, which is the focus of this paper. Unfortunately, two key ingredients that help deep networks attain high accuracy — over-parameterization, and the randomization of their training algorithms — pose significant challenges to their reproducibility. The former refers to the fact that NNs typically have many solutions that minimize the training objective (Neyshabur et al., 2015; Zhang et al., 2017). The latter refers to the fact that standard training of NNs involves several sources of randomness, e.g., initialization, mini-batch ordering, non-determinism in training platforms and in some cases data augmentation. Put together, these imply that NN training can find vastly different solutions in each run even when training data is the same, leading to a reproducibility challenge.

The prediction disagreement between two models is referred to as *churn* (Cormier et al., 2016)[1]. Concretely, given two models, churn is the fraction of test examples where the predictions of the two models disagree. Clearly, churn is zero if both models have perfect accuracy – an unattainable goal for most of the practical settings of interest. Similarly, one can mitigate churn by eliminating all sources of randomness in the underlying training setup. However, even if one controls the seed used for random initialization and the order of data, inherent non-determinism in the current computation platforms is hard to avoid (see §2.3). Moreover it is desirable to have stable models with predictions

---

[1] Madani et al. (2004) referred to this as disagreement and used it as an estimate for generalization error and model selection

unaffected by such factors in training. Thus, it is critical to quantify churn, and develop methods that reduce it.

In this paper, we study the problem of churn in NNs for the *classification setting*. We demonstrate the presence of churn, and investigate the role of different training factors causing it. Interestingly our experiments show that churn is not avoidable on the computing platforms commonly used in machine learning, further highlighting the necessity of developing techniques to mitigate churn. We then analyze the relation between churn and predicted class probabilities. Based on this, we develop a novel regularized co-distillation approach for reducing churn.

Our key contributions are summarized below:

 (i) Besides the disagreement in the final predictions of models, we propose alternative soft metrics to measure churn. We demonstrate the existence of churn on standard image classification tasks (CIFAR-10, CIFAR-100, ImageNet, SVHN and iNaturalist), and identify the components of learning algorithms that contribute to the observed churn. Furthermore, we analyze the relationship between churn and model prediction confidences (cf. § 2).

 (ii) Motivated from our analysis, we propose a regularized co-distillation approach to reduce churn that both improves prediction confidences and reduces prediction variance (cf. §3). Our approach consists of two components: a) minimum entropy regularizers that improve prediction confidences (cf. §3.1), and b) a new variant of co-distillation (Anil et al., 2018) to reduce prediction variance across runs. Specifically, we use a symmetric KL divergence based loss to reduce model disagreement, with a linear warmup and joint updates across multiple models (cf. §3.2).

(iii) We empirically demonstrate the effectiveness of the proposed approach in reducing churn and (sometimes) increasing accuracy. We present ablation studies over its two components to show their complementary nature in reducing churn (cf. §4).

## 1.1 RELATED WORK

**Reproducibility in machine learning.** There is a broad field studying the problem of *reproducible research* (Buckheit & Donoho, 1995; Gentleman & Lang, 2007; Sonnenburg et al., 2007; Kovacevic, 2007; Mesirov, 2010; Peng, 2011; McNutt, 2014; Braun & Ong, 2014; Rule et al., 2018), which identifies best practices to facilitate the reproducibility of scientific results. Henderson et al. (2018) analysed reproducibility of methods in reinforcement learning, showing that performance of certain methods is sensitive to the random seed used in the training. While the performance of NNs on image classification tasks is fairly stable (Table 2), we focus on analyzing and improving the reproducibility of individual predictions. Thus, churn can be seen as a specific technical component of this reproducibility challenge.

Cormier et al. (2016) defined the disagreement between predictions of two models as churn. They proposed an MCMC approach to train an initial stable model A so that it has a small churn with its future version, say model B. Here, future versions are based on slightly modified training data with possibly additional features. In Goh et al. (2016); Cotter et al. (2019), constrained optimization is utilized to reduce churn across different model versions. In contrast, we are interested in capturing the contribution of factors other than training data modification that cause churn.

More recently. Madhyastha & Jain (2019) study instability in the interpretation mechanisms and average performance for deep NNs due to change in random seed, and propose a stochastic weight averaging (Izmailov et al., 2018) approach to promote robust interpretations. In contrast, we are interested in robustness of individual predictions.

**Ensembling and online distillation.** Ensemble methods (Dietterich, 2000; Lakshminarayanan et al., 2017) that combine the predictions from multiple (diverse) models naturally reduce the churn by averaging out the randomness in the training procedure of the individual models. However, such methods incur large memory footprint and high computational cost during the inference time. Distillation (Hinton et al., 2015; Bucilua et al., 2006) aims to train a single model from the ensemble to alleviate these costs. Even though distilled model aims to recover the accuracy of the underlying ensemble, it is unclear if the distilled model also leads to churn reduction. Furthermore, distillation is a two-stage process, involving first training an ensemble and then distilling it into a single model.

To avoid this two-stage training process, multiple recent works Anil et al. (2018); Zhang et al. (2018); Lan et al. (2018); Song & Chai (2018); Guo et al. (2020) have focused on *online distillation*, where

multiple identical or similar models (with different initialization) are trained while regularizing the distance between their prediction probabilities. At the end of the training, any of the participating models can be used for inference. Notably, Anil et al. (2018), while referring to this approach as co-distillation, also empirically pointed out its utility for churn reduction on the Criteo Ad dataset[2]. In contrast, we develop a deeper understanding of co-distillation framework as a churn reduction mechanism by providing a theoretical justification behind its ability to reduce churn. We experimentally show that using a symmetric KL divergence objective instead of the cross entropy loss for co-distillation (Anil et al., 2018) leads to lower churn and better accuracy, even improving over the expensive ensembling-distillation approach.

**Entropy regularizer.** Minimum entropy regularization was earlier explored in the context of semi-supervised learning (Grandvalet & Bengio, 2005). Such techniques have also been used to combat label noise (Reed et al., 2015). In contrast, we utilize minimum entropy regularization in fully supervised settings for a distinct purpose of reducing churn and experimentally show its effectiveness.

## 1.2 NOTATION

**Multi-class classification.** We consider a multi-class classification setting, where given an instance $x \in \mathcal{X}$, the goal is to classify it as a member of one of $K$ classes, indexed by the set $\mathcal{Y} \triangleq [K]$. Let $\mathcal{W}$ be the set of parameters that define the underlying classification models. In particular, for $w \in \mathcal{W}$, the associated classification model $f(\cdot; w) \colon \mathcal{X} \to \Delta_K$ maps the instance $x \in \mathcal{X}$ in the $K$-dimensional simplex $\Delta_K \subset \mathbb{R}^K$. Given $f(x; w)$, $x$ is classified as element of class $\hat{y}_{x;w}$ such that

$$\hat{y}_{x;w} = \arg\max_{j \in \mathcal{Y}} f(x; w)_j. \tag{1}$$

This gives the misclassification error $\ell_{01}(y, \hat{y}_{x;w}) = \mathbb{1}_{\{\hat{y}_{x;w} \neq y\}}$, where $y$ is the true label for $x$. Let $\mathbb{P}_{X,Y}$ be the joint distribution over the instance and label pairs. We learn a classification model by minimizing the risk for some valid surrogate loss $\ell$ of the misclassification error $\ell_{01}$: $L(w) \triangleq \mathbb{E}_{X,Y}[\ell(Y, f(X; w))]$. In practice, since we have only finite samples $\mathcal{S} \in (\mathcal{X} \times \mathcal{Y})^n$, we minimize the corresponding empirical risk.

$$L(w; \mathcal{S}) \triangleq \frac{1}{|\mathcal{S}|} \sum_{(x,y) \in \mathcal{S}} \ell(y, f(x; w)). \tag{2}$$

## 2 CHURN: MEASUREMENT AND ANALYSIS

In this section we define churn, and demonstrate its existence on CIFAR and ImageNet datasets. We also propose and measure alternative soft metrics to quantify churn, that mitigate the discontinuity of churn. Subsequently, we examine the influence of different factors in the learning algorithm on churn. Finally, we present a relation between churn and prediction confidences of the model.

We begin by defining churn as the expected disagreement between the predictions of two models (Cormier et al., 2016).

**Definition 1** (Churn between two models). Let $w_1, w_2 \in \mathcal{W}$ define classification models $f(\cdot; w_1), f(\cdot; w_2) \colon \mathcal{X} \to \Delta^K$, respectively. Then, the churn between the two models is

$$\mathrm{Churn}(w_1, w_2) = \mathbb{E}_X[\mathbb{1}_{\{\hat{Y}_{x;w_1} \neq \hat{Y}_{x;w_2}\}}] = \mathbb{P}_X[\hat{Y}_{X;w_1} \neq \hat{Y}_{X;w_2}], \tag{3}$$

where $\hat{Y}_{x;w_1} \triangleq \arg\max_{j \in \mathcal{Y}} f(x; w_1)_j$ and $\hat{Y}_{x;w_2} \triangleq \arg\max_{j \in \mathcal{Y}} f(x; w_2)_j$.

Note that if the models have perfect test accuracy, then their predictions always agree with the true label, which corresponds to zero churn. In practice, however, this is rarely the case. The following rather straightforward result shows that churn is upper bounded by the sum of the test error of the models. See Appendix B for the proof. We note that a similar result was shown in Theorem 1 of Madani et al. (2004).

**Lemma 1.** *Let* $\mathrm{P}_{\mathrm{Err},w_1} \triangleq \mathbb{P}_{X,Y}[Y \neq \hat{Y}_{X;w_1}]$ *and* $\mathrm{P}_{\mathrm{Err},w_2} \triangleq \mathbb{P}_{X,Y}[Y \neq \hat{Y}_{X;w_2}]$ *be the misclassification error for the models* $w_1$ *and* $w_2$, *respectively. Then,* $\mathrm{Churn}(w_1, w_2) \leqslant \mathrm{P}_{\mathrm{Err},w_1} + \mathrm{P}_{\mathrm{Err},w_2}$.

---

[2]https://www.kaggle.com/c/criteo-display-ad-challenge

| | | | | | | | | |
|---|---|---|---|---|---|---|---|---|
| Remove augmentation | | | | | ✓ | ✓ | ✓ | ✓ |
| Identical minibatch order | | | ✓ | ✓ | | | ✓ | ✓ |
| Identical initialization | | ✓ | | ✓ | | ✓ | | ✓ |
| Accuracy | $94.02 \pm 0.09$ | $94.00 \pm 0.18$ | $93.88 \pm 0.17$ | $93.86 \pm 0.24$ | $89.24 \pm 0.06$ | $89.00 \pm 0.16$ | $87.67 \pm 0.25$ | $87.05 \pm 0.01$ |
| Churn(%) | $5.81 \pm 0.08$ | $5.56 \pm 0.12$ | $5.96 \pm 0.26$ | $5.79 \pm 0.18$ | $11.08 \pm 0.10$ | $11.20 \pm 0.23$ | $12.36 \pm 0.16$ | $12.20 \pm 0.29$ |
| $\text{SChurn}_1$(%) | $6.55 \pm 0.12$ | $6.32 \pm 0.09$ | $6.77 \pm 0.18$ | $6.56 \pm 0.17$ | $13.07 \pm 0.13$ | $13.04 \pm 0.24$ | $14.25 \pm 0.14$ | $14.11 \pm 0.17$ |

Table 1: Ablation study of churn across 5 runs on CIFAR-10 with a ResNet-56. Holding the initialization constant across models always decreases churn, but using identical mini-batch ordering and completely removing augmentation can increase churn with a decrease in accuracy.

Despite the worst-case bound in Lemma 1, imperfect accuracy does not preclude the absence of churn. As the best-case scenario, two imperfect models can agree on the predictions for each example (whether correct or incorrect), causing the churn to be zero. For example, multiple runs of a deterministic learning algorithm produce models with zero churn, independent of their accuracy.

This shows that, in general, one cannot infer churn from test accuracy, and understanding churn of an algorithm requires independent exploration.

### 2.1 DEMONSTRATION OF CHURN

In principle, there are multiple sources of randomness in standard training procedures that make NNs susceptible to churn. We now verify this hypothesis by showing that, in practice, these sources indeed result in a non-trivial churn for NNs on standard image classification datasets.

Table 2 reports churn over CIFAR-10, CIFAR-100 and ImageNet, with ResNets (He et al., 2016a;b) as the underlying model architecture. As per the standard practice in the literature, we employ SGD with momentum and step-wise learning rate decay to train ResNets. Note that the models obtained in different runs indeed disagree in their predictions substantially. To measure if the disagreement comes mainly from misclassified examples, we measure churn on two slices of examples, correctly classified (Churn correct) and misclassified (Churn incorrect). We notice churn even among the examples that are correctly classified, though we see a relatively higher churn among the misclassified examples.

This raises two natural questions: (i) do the prediction probabilities differ significantly, or is the churn observed in Table 2 mainly an artifact of the $\mathrm{argmax}$ operation in (1), when faced with small variation in prediction probabilities across models?, and (ii) what causes such a high churn across runs? We address these questions in the following sections.

### 2.2 SURROGATE CHURN

Churn in Table 2 could potentially be a manifestation of applying the $\arg\max$ operation in (1), despite the prediction probabilities being close. To study this, we consider the following soft metric to measure churn, which takes the models' entire prediction probability mass function into account.

**Definition 2** (Surrogate churn between two models). Let $f(\cdot; w_1), f(\cdot; w_2) \colon \mathcal{X} \to \Delta^K$ be two models defined by $w_1, w_2 \in \mathcal{W}$, respectively. Then, for $\alpha \in \mathbb{R}^+$, the surrogate churn between the models is

$$\text{SChurn}_\alpha(w_1, w_2) = \frac{1}{2} \cdot \mathbb{E}_{\mathsf{X}} \left[ \left\| \left( \frac{f(\mathsf{X}; w_1)}{\max f(\mathsf{X}; w_1)} \right)^\alpha - \left( \frac{f(\mathsf{X}; w_2)}{\max f(\mathsf{X}; w_2)} \right)^\alpha \right\|_1 \right]. \quad (4)$$

As $\alpha \to \infty$, this *reduces* to the standard Churn definition in (3). In Table 2 we measure $\text{SChurn}_\alpha$ for $\alpha = 1$, which shows that even the distance between prediction probabilities is significant across runs. Thus, the churn observed in Table 2 is not merely caused by the discontinuity of the $\arg\max$, but it indeed highlights the instability of model predictions caused by randomness in training.

### 2.3 WHAT CAUSES CHURN?

We now investigate the role played by randomness in initialization, mini-batch ordering, data augmentation, and non-determinism in the computation platform in causing churn. Even though these aspects are sources of *randomness* in training, they are not necessarily sources of *churn*. We experiment by holding some of these factors constant and measure the churn across 5 different runs. We report results in Table 1, where the first column gives the baseline churn with no factor held constant across runs.

**Model initialization** is a source of churn. Simply initializing weights from identical seeds (odd columns in Table 1) can decrease churn under most settings. Other sources of randomness include **mini-batch ordering** and **dataset augmentation.** To hold the former constant, we ensure that every model iterates over the dataset in the same order; to hold the latter constant, we remove all augmentation during training. These two aspects contribute to randomness between training runs, but fixing them does not decrease churn; rather, they appear to have a regularizing affect on our hardware platform.

Finally, there is churn resulting from an unavoidable source during training, the non-determinism in **computing platforms** used for training machine learning models, e.g., GPU/TPU (Morin & Willetts, 2020; PyTorch, 2019; Nvidia, 2020). The experiments in Table 1 were run on TPU. Even when all other aspects of training are held constant (rightmost column), model weights diverge within 100 steps (across runs) and the final churn is significant. We verified that this is the sole source of non-determinism: models trained to 10,000 steps on CPUs under these settings had identical weights.

These experiments underscore the importance of developing and incorporating churn reduction strategies in training. Even with extreme measures to eliminate all sources of randomness, we continue to observe churn due to unavoidable hardware non-determinism.

## 2.4 REDUCING CHURN: RELATION TO PREDICTION PROBABILITIES

We now focus on the main objective of this paper – churn reduction in NNs. Many factors that have been shown to cause churn in § 2.3 are crucial for the good performance of NNs and thus cannot be controlled or eliminated without causing performance degradation. Moreover, controlling certain factors such as hardware non-determinism is extremely challenging, especially in the large scale distributed computing setup.

Towards reducing churn, we first develop an understanding of the relationship between the distribution of prediction probabilities of a model and churn. Intuitively, encouraging either larger prediction confidence or smaller variance across multiple training runs should result in reduced churn.

To formalize this intuition, let us consider the *prediction confidence* realized by the classification model $f(\cdot; w)$ on the instance $x \in \mathcal{X}$:

$$\gamma_{x;w} = f(x; w)_{\hat{y}_{x;w}} - \arg\max_{j \neq \hat{y}_{x;w}} f(x; w)_j, \tag{5}$$

where $\hat{y}_{x;w}$ is the model prediction in (1). Note that $\gamma_{x;w}$ denotes the difference between the probabilities of the most likely and the second most likely classes under the distribution $f(x; w)$, and is *not* the same as the standard multi-class margin (Koltchinskii et al., 2001): it captures how confident the model $f(\cdot; w)$ is about its prediction $\hat{y}_{x;w}$, without taking the *true* label into account.

The following result relates the prediction confidence to churn. See Appendix B for the proof.

**Lemma 2.** *Let $\gamma_{x;w_1}$ and $\gamma_{x;w_2}$ be the prediction confidences realized on $x \in \mathcal{X}$ by the classification models $f(\cdot; w_1)$ and $f(\cdot; w_2)$, respectively. Then,*

$$\text{Churn}(w_1, w_2) = \mathbb{P}_{\mathsf{X}}\{\hat{\mathsf{Y}}_{\mathsf{X};w_1} \neq \hat{\mathsf{Y}}_{\mathsf{X};w_2}\} \leqslant \mathbb{P}_{\mathsf{X}}\big[D_{\text{L1}}(f(\mathsf{X}; w_1), f(\mathsf{X}; w_2)) > \min\{\gamma_{\mathsf{X};w_1}, \gamma_{\mathsf{X};w_2}\}\big].$$

Here, $D_{\text{L1}}(f(x; w_1), f(x; w_2)) = \sum_{j \in \mathcal{Y}} |f(x; w_1)_j - f(x; w_2)_j|$ measures the $L1$ distance. Lemma 2 establishes that, for a given instance $x$, the churn between two models $f(\cdot; w_1)$ and $f(\cdot; w_2)$ becomes less likely as their confidences $\gamma_{x;w_1}$ and $\gamma_{x;w_2}$ increase. Similarly, the churn becomes smaller when the difference between their prediction probabilities, $D_{\text{L1}}(f(x; w_1), f(x; w_2))$, decreases.

## 3 REGULARIZED CO-DISTILLATION FOR CHURN REDUCTION

In this section we present our approach for churn reduction. Motivated by Lemma 2, we consider an approach with two *complementary* techniques for churn reduction: 1) We first propose entropy based regularizers that encourage solutions $w \in \mathcal{W}$ with large prediction confidence $\gamma_{x;w}$ for each instance $x$. 2) We next employ co-distillation approach with novel design choices that simultaneously trains two models while minimizing the distance between their prediction probabilities.

Note that the entropy regularizers themselves cannot *actively* enforce alignment between the prediction probabilities across multiple runs as the resulting objective does not couple multiple models together.

On the other hand, the co-distillation in itself does not affect the prediction confidence of the underlying models (as verified in Figure 2). Thus, our combined approach promotes both large model prediction confidence and small variance in prediction probabilities across multiple runs.

### 3.1 MINIMUM ENTROPY REGULARIZERS

Aiming to increase the prediction confidences of the trained model $\{\gamma_{x;w}\}_{x\in\mathcal{X}}$, we propose novel training objectives that employ one of two possible regularizers based on: (1) entropy of the model prediction probabilities; and (2) negative symmetric KL divergence between the model prediction probabilities and the uniform distribution. Both regularizers encourage the prediction probabilities to be concentrated on a small number of classes, and increase the associated prediction confidence.

**Entropy regularizer.** Recall that the standard training minimizes the risk $L(w)$. Instead, for $\alpha \in [0,1]$ and $\mathcal{S} = \{(x_i, y_i)\}_{i\in[n]}$, we propose to minimize the following regularized objective to increase the prediction confidence of the model[3].

$$L_{\text{entropy}}(w; \mathcal{S}) = (1 - \alpha) \cdot L(w; \mathcal{S}) + \alpha \cdot \frac{1}{n} \sum\nolimits_{i\in[n]} H\big(f(x_i, w)\big), \tag{6}$$

where $H\big(f(x; w)\big) = -\sum_{j\in[K]} f(x; w)_j \log f(x; w)_j$ denotes the entropy of predictions $f(x; w)$.

**Symmetric KL divergence regularizer.** Instead of encouraging the low entropy for the prediction probabilities, we can alternatively maximize the distance of the prediction probability mass function from the uniform distribution as a regularizer to enhance the prediction confidence. In particular, we utilize the symmetric KL divergence as the distance measure. Let $\text{Unif} \in \Delta_K$ be the uniform distribution. Thus, given $n$ samples $\mathcal{S} = \{(x_i, y_i)\}_{i\in[n]}$, we propose to minimize

$$L_{\text{SKL}}(w; \mathcal{S}) \triangleq (1 - \alpha) \cdot L(w; \mathcal{S}) - \alpha \cdot \frac{1}{n} \sum\nolimits_{i\in[n]} \text{SKL}\big(f(x_i; w), \text{Unif}\big), \tag{7}$$

where $\text{SKL}\big(f(x; w), \text{Unif}\big) = \text{KL}\big(f(x; w) || \text{Unif}\big) + \text{KL}\big(\text{Unif} || f(x; w)\big)$.

As discussed in the beginning of the section, the intuition behind utilizing these regularizers is to encourage spiky prediction probability mass functions, which lead to higher prediction confidence. The following result supports this intuition in the binary classification setting. See Appendix B for the proof. As for multi-class classification, we empirically verify that the proposed regularizers indeed lead to increased prediction confidences (cf. Figure 2).

**Theorem 3.** *Let $f(\cdot; w)$ and $f(\cdot; w')$ be two binary classification models. For a given $x \in \mathcal{X}$, if we have $H(f(x; w)) \leqslant H(f(x; w'))$ or $\text{SKL}(f(x; w)) \geqslant \text{SKL}(f(x; w'))$, then $\gamma_{x;w} \geqslant \gamma_{x;w'}$.*

Note that the effect of these regularizers is different from increasing the temperature in softmax while computing $f(x; w)$. Similar to max entropy regularizers (Pereyra et al., 2017), they are independent of the label, a crucial difference that allows them to reduce churn even among misclassified examples.

### 3.2 CO-DISTILLATION

As the second measure for churn reduction, we now focus on designing training objectives that minimize the variance among the prediction probabilities for the same instance across multiple runs of the learning algorithm. Towards this, we consider novel variants of co-distillation approach (Anil et al., 2018). In particular, we employ co-distillation by using symmetric KL divergence (Co-distill$_{\text{SKL}}$), with a linear warm-up, to penalize the prediction distance between the models instead of the popular step-wise cross entropy loss (Co-distill$_{\text{CE}}$) used in Anil et al. (2018) (see § C). We first motivate the proposed objective for reducing churn using Lemma 2.

Recall from Lemma 2 that, for a given instance $x$, churn across two models $f(\cdot; w_1)$ and $f(\cdot; w_2)$ decreases as the distance between their prediction probabilities $D_{\text{L1}}(f(x; w_1), f(x; w_2))$ becomes smaller. Motivated by this, given training samples $\mathcal{S} = \{(x_i, y_i)\}_{i\in[n]}$, one can simultaneously train two models corresponding to $w_1, w_2 \in \mathcal{W}$ by minimizing the following objective and *keeping either*

---

[3]We restrict ourselves to the convex combination of the original risk and the regularizer terms as this ensures that the scale of the proposed objective is the same as that of the original risk. This allows us to experiment with the same learning rate and other hyperparameters for both.

| Dataset | Method | Training cost | Accuracy | Churn(%) | SChurn$_1$(%) | Churn correct | Churn incorrect |
|---|---|---|---|---|---|---|---|
| CIFAR-10 ResNet-56 | Baseline | 1x | 93.97±0.11 | 5.72±0.18 | 6.41±0.15 | 2.62±0.12 | 53.82±1.44 |
| | Entropy (this paper) | 1x | 94.05±0.18 | 5.59±0.21 | 6.17±0.17 | 2.55±0.19 | 53.33±1.62 |
| | SKL (this paper) | 1x | 93.75±0.13 | 5.79±0.18 | 6.03±0.16 | 2.66±0.15 | 52.6±1.59 |
| | 2-Ensemble distil. (Hinton et al., 2015) | 3x | 94.62±0.07 | 4.47±0.13 | 4.91±0.11 | 2.05±0.1 | 47.25±1.55 |
| | Co-distill$_{CE}$ (Anil et al., 2018) | 2x | 94.25±0.15 | 5.14±0.14 | 5.62±0.12 | 3.03±0.31 | 40.45±5.98 |
| | Co-distill$_{SKL}$ (this paper) | 2x | 94.63±0.15 | 4.29±0.14 | 4.66±0.11 | 2.49±0.25 | **36.78±4.14** |
| | +Entropy (this paper) | 2x | **94.63±0.15** | **4.21±0.15** | **4.61±0.14** | **1.94±0.13** | 44.29±1.9 |
| CIFAR-100 ResNet-56 | Baseline | 1x | 73.26±0.22 | 26.77±0.26 | 37.19±0.35 | 11.42±0.26 | 68.77±0.72 |
| | Entropy (this paper) | 1x | 73.24±0.29 | 26.55±0.26 | 34.58±0.29 | 11.29±0.32 | 68.32±0.69 |
| | SKL (this paper) | 1x | 73.35±0.3 | 25.62±0.36 | 28.81±0.35 | 11.16±0.34 | 65.42±0.65 |
| | 2-Ensemble distil. (Hinton et al., 2015) | 3x | 76.27±0.25 | 18.86±0.26 | 25.43±0.23 | 7.79±0.23 | 54.3±0.86 |
| | Co-distill$_{CE}$ (Anil et al., 2018) | 2x | 75.11±0.17 | 19.55±0.27 | 25.03±0.27 | 9.98±0.97 | 48.69±2.84 |
| | Co-distill$_{SKL}$ (this paper) | 2x | **76.58±0.16** | 17.70±0.33 | 24.99±0.31 | 9.08±0.8 | **45.98±2.98** |
| | +Entropy (this paper) | 2x | 76.53±0.26 | **17.09±0.3** | **24.06±0.28** | **7.2±0.26** | 49.4±1.1 |
| ImageNet ResNet-v2-50 | Baseline | 1x | 76±0.11 | 15.22±0.16 | 35.01±0.24 | 5.87±0.11 | 44.77±0.46 |
| | Entropy (this paper) | 1x | 76.39±0.11 | 14.92±0.09 | 32.93±0.18 | 5.75±0.11 | 44.56±0.33 |
| | SKL (this paper) | 1x | 75.98±0.08 | 15.32±0.11 | 32.64±0.2 | 5.99±0.07 | 44.88±0.37 |
| | 2-Ensemble distil. (Hinton et al., 2015) | 3x | 75.83±0.08 | 12.92±0.14 | 34.02±0.2 | 4.92±0.11 | 38.04±0.39 |
| | Co-distill$_{CE}$ (Anil et al., 2018) | 2x | 76.12±0.08 | 11.62±0.21 | **18.38±0.34** | 4.97±0.36 | 32.67±1.45 |
| | Co-distill$_{SKL}$ (this paper) | 2x | **76.74±0.06** | 11.45±0.13 | 28.66±0.20 | 5.11±0.36 | **32.41±1.15** |
| | +Entropy (this paper) | 2x | 76.56±0.02 | **11.05±0.36** | 22.92±1.01 | **4.32±0.32** | 32.89±1.15 |
| iNaturalist ResNet-v2-50 | Baseline | 1x | 60.97±0.26 | 26.99±0.25 | 71.33±0.51 | 9.45±0.3 | 54.28±0.5 |
| | Entropy (this paper) | 1x | **62.74±0.15** | 25.84±0.2 | **66.61±0.47** | 9.23±0.24 | 53.82±0.35 |
| | Co-distill$_{CE}$ (Anil et al., 2018) | 2x | 61.1±0.11 | 26.89±0.22 | 70.81±0.43 | 9.51±0.22 | 54.18±0.41 |
| | Co-distill$_{SKL}$ (this paper) | 2x | 61.59±0.28 | **21.73±0.26** | 68.11±1.17 | **7.48±0.34** | **44.53±0.59** |
| SVHN LeNet5 | Baseline | 1x | 87.16±0.43 | 14.3±0.4 | 17.52±0.48 | 5.9±0.33 | 70.39±1.24 |
| | Entropy (this paper) | 1x | 87.24±0.61 | 14.12±0.42 | 16.62±0.52 | 6.1±0.47 | 69.31±1.41 |
| | Co-distill$_{CE}$ (Anil et al., 2018) | 2x | 88.93±0.39 | 9.42±0.33 | **10.44±0.36** | 3.84±0.37 | 54.09±1.83 |
| | Co-distill$_{SKL}$ (this paper) | 2x | **89.61±0.25** | **8.49±0.25** | 12.2±0.35 | **3.59±0.24** | **51.11±1.46** |

Table 2: Estimate of Churn (cf. (3)) and SChurn (cf. (4)) on the test sets. For each setting, we report the mean and standard deviation over 10 independent runs, with random initialization, mini-batches and data-augmentation. We report the values corresponding to the smallest churn for each method (see Table 5 in § A for the exact parameters). We boldface the best results in each column. First we notice that both the proposed methods are effective at reducing churn and Schurn, with Co-distill$_{SKL}$ showing significant reduction in churn. Additionally, these methods also improve the accuracy. Finally combing the entropy regularizer with Co-distill$_{SKL}$ offers the best way to reduce churn, improving over the ensembling-distillation and Co-distill$_{CE}$ approaches. Note that for co-distillation we measure churn of a single model (e.g. $f(;w_1)$ in eq (9)) across independent training runs.

*of the two models* as the final solution.

$$L_{\text{Co-distill}_{L1}}(w_1, w_2; \mathcal{S}) \triangleq L(w_1; \mathcal{S}) + L(w_2; \mathcal{S}) + \frac{\beta}{n} \sum_{i \in [n]} D_{L1}(f(x_i; w_1), f(x_i; w_2)) \quad (8)$$

From Pinsker's inequality, $D_{L1}(f(x_i; w_1), f(x_i; w_2)) \leqslant \sqrt{2 \cdot \text{KL}(f(x_i; w_1) || f(x_i; w_2))}$. Thus, one can alternatively utilize the following objective.

$$L_{\text{Co-distill}_{SKL}}(w_1, w_2; \mathcal{S}) \triangleq L(w_1; \mathcal{S}) + L(w_2; \mathcal{S}) + \frac{\beta}{n} \sum_{i \in [n]} \text{SKL}(f(x_i; w_1), f(x_i; w_2)), \quad (9)$$

where $\text{SKL}(f(x; w_1), f(x; w_2)) = \text{KL}(f(x; w_1) || f(x; w_2)) + \text{KL}(f(x; w_2) || f(x; w_1))$ denotes the symmetric KL-divergence between the prediction probabilities of the two models. In what follows, we work with the objective in (9) as we observed this to be more effective in our experiments, leading to both smaller churn and higher model accuracy.

In addition to the co-distillation objective, we introduce two other changes to the training procedure: joint updates and linear rampup of the co-distillation loss. We discuss these differences in § C.

**Regularized co-distillation.** In this paper, we also explore *regularized* co-distillation, where we utilize the minimum entropy regularizes (cf. § 3.1) in our co-distillation framework. We note that the best results for the combined approach are achieved when we use a linear warmup for the regularizer coefficient as well. Note that combining the Co-distill$_{SKL}$ objective with an entropy regularizer is not the same as using the cross entropy loss, due to the use the different *weights* for each method, and rampup of the regularizer coefficients. This distinction is important in reducing churn ( Table 2).

## 4 EXPERIMENTS

We conduct experiments on 5 different datasets, CIFAR-10, CIFAR-100, ImageNet, SVHN and iNaturalist 2018. We use LeNet5 for experiments on SVHN, ResNet-56 for experiments on CIFAR-10 and CIFAR-100, and ResNet-v2-50 for experiments on ImageNet and iNaturalist. We use the

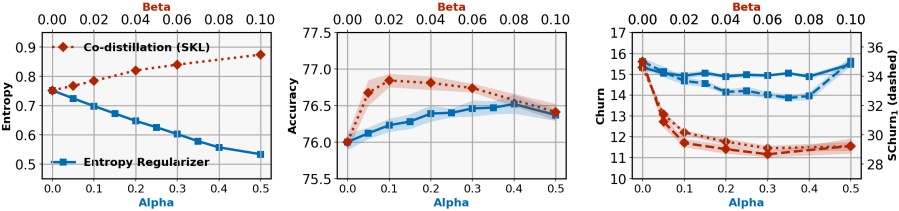

Figure 1: ImageNet ablation study: We plot the effect of the entropy regularizer (top$_{10}$), for varying $\alpha$, and Co-distill$_{SKL}$ for varying $\beta$, on the prediction entropy, accuracy and churn. These plots shows the *complementary* nature of these methods in reducing churn. While entropy regularizer reduces churn by reducing the prediction entropy, Co-distill$_{SKL}$ reduces churn by improving the agreement between two models, hence increasing the entropy.

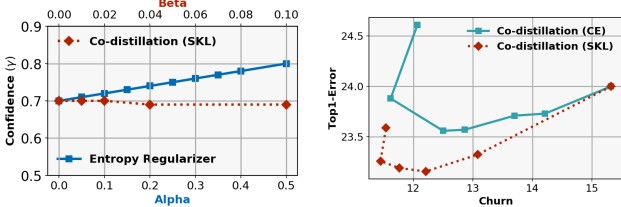

Figure 2: ImageNet: Left - effect of the entropy regularizer and our proposed variant of co-distillation, Co-distill$_{SKL}$, on the prediction confidence. Right - comparison with Co-distill$_{CE}$ (Anil et al., 2018). We plot the trade-off between accuracy and churn for Co-distill$_{CE}$ and Co-distill$_{SKL}$ by varying $\beta$. The plot clearly shows that the proposed variant achieves a better tradeoff, compared to Co-distill$_{CE}$.

| Dataset | Method | Accuracy | | Churn(%) | |
|---|---|---|---|---|---|
| | | Weight decay = 0 | Weight decay = 0.0001 | Weight decay = 0 | Weight decay = 0.0001 |
| CIFAR-10 | Baseline | 91.62$\pm$0.2 | 93.97$\pm$0.11 | 8.4$\pm$0.21 | 5.72$\pm$0.18 |
| | Entropy (this paper) | 91.56$\pm$0.26 | 94.05$\pm$0.18 | 8.45$\pm$0.25 | 5.59$\pm$0.21 |
| | SKL (this paper) | 91.91$\pm$0.22 | 93.75$\pm$0.13 | 7.84$\pm$0.24 | 5.79$\pm$0.18 |
| | Co-distill$_{SKL}$ (this paper) | **92.1$\pm$0.31** | **94.63$\pm$0.15** | **6.74$\pm$0.32** | **4.29$\pm$0.14** |
| CIFAR-100 | Baseline | 68.6$\pm$0.21 | 73.26$\pm$0.22 | 32.23$\pm$0.45 | 26.77$\pm$0.26 |
| | Entropy (this paper) | 68.41$\pm$0.33 | 73.24$\pm$0.29 | 32.02$\pm$0.42 | 26.55$\pm$0.26 |
| | SKL (this paper) | 69.02$\pm$0.33 | 73.35$\pm$0.3 | 31.22$\pm$0.38 | 25.62$\pm$0.36 |
| | Co-distill$_{SKL}$ (this paper) | **72.29$\pm$0.42** | **76.58$\pm$0.16** | **23.14$\pm$0.83** | **17.70$\pm$0.33** |

Table 3: Weight decay ablation studies. Similar to Table 2, we provide results on accuracy and churn for baseline and the proposed approaches, with and without weight decay. We notice that the proposed approaches improve churn both with and without weight decay.

same hyperparameters for all the experiments on a dataset. We use the Cross entropy loss and the Momentum optimizer. For complete details we refer to the Appendix A.

**Top$_k$ regularizer.** For problems with a large number of outputs, e.g. ImageNet, it is not required to penalize all the predictions to reduce churn. Recall that prediction confidence is only a function of the top two prediction probabilities. Hence, on ImageNet, we consider the top$_k$ variants of the proposed regularizers, and penalize the entropy/SKL only on the top $k$ predictions, with $k = 10$.

**Accuracy and churn.** In Table 2, we present the accuracy and churn of models trained with the minimum entropy regularizers, co-distillation and their combination. For each dataset we report the values for the best $\alpha$ and $\beta$. We notice that our proposed methods indeed reduce the churn and Schurn. While both minimum entropy regularization and co-distillation are consistently effective in reducing Schurn, their effect on churn varies, potentially due to the discontinuous nature of the churn. Also note that the proposed methods reduce churn both among the correctly and incorrectly classified examples. Our co-distillation proposal, Co-distill$_{SKL}$ consistently showed a significant reduction in churn, and has better performance than Co-distill$_{CE}$ (Anil et al., 2018) that is based on the cross-entropy loss, showing the importance of choosing the right objective for reducing churn. We present additional comparison in Figure 2, showing churn and Top-1 error on ImageNet, computed for

| Method | Training cost | CIFAR-10 - ECE ($\times 1e2$) | CIFAR-100 - ECE ($\times 1e2$) | ImageNet - ECE ($\times 1e2$) |
|---|---|---|---|---|
| Baseline | 1x | $4.04 \pm 0.13$ | $15.23 \pm 0.32$ | $3.71 \pm 0.12$ |
| Entropy (this paper) | 1x | $4.19 \pm 0.18$ | $17.43 \pm 0.26$ | $6.17 \pm 0.09$ |
| SKL (this paper) | 1x | $5.41 \pm 0.12$ | $21.82 \pm 0.16$ | $6.49 \pm 0.12$ |
| 2-Ensemble distil. (Hinton et al., 2015) | 3x | $\mathbf{3.83 \pm 0.05}$ | $13.85 \pm 0.23$ | $\mathbf{2.14 \pm 0.1}$ |
| Co-distill$_{\mathrm{CE}}$ (Anil et al., 2018) | 2x | $4.13 \pm 0.12$ | $15.86 \pm 0.13$ | $11.97 \pm 0.07$ |
| Co-distill$_{\mathrm{SKL}}$ (this paper) | 2x | $3.88 \pm 0.15$ | $\mathbf{11.39 \pm 0.15}$ | $2.44 \pm 0.09$ |
| +Entropy (this paper) | 2x | $3.94 \pm 0.09$ | $11.55 \pm 0.23$ | $3.89 \pm 0.39$ |

Table 4: **Expected Calibration Error.** We compute the Expected Calibration Error (ECE) (Guo et al., 2017) to evaluate the effect on calibration of logits by the churn reduction methods considered in this paper, for different datasets. We report ECE for the predictions of the models used to report accuracy and churn in Table 2. We note that while the minimum entropy regularizers, predictably, increase the calibration error, our combined approach with co-distillation results in calibration error competitive with 2-ensemble distillation method.

different values of $\beta$ (cf. (9)). Finally, we achieve an even further reduction in churn by the combined approach of entropy regularized co-distillation. Interestingly, the considered methods also improve the accuracy, showing their utility beyond churn reduction.

**Ensembling.** We also compare with the ensembling-distillation approach (Hinton et al., 2015) in Table 2, where we use a 2 teacher ensemble for distillation. We show that the proposed methods consistently outperform ensembling-distillation approach, despite having a lower training cost.

**Ablation.** We next report results of our ablation study on the entropy ($\mathrm{top}_{10}$) regularizer coefficient $\alpha$, and the Co-distill$_{\mathrm{SKL}}$ coefficient $\beta$ in Figure 1. While both methods improve accuracy and churn, the prediction entropy shows different behavior between these methods. While entropy regularizers improve churn by reducing the entropy, co-distillation reduces churn by reducing the prediction distance between two models, resulting in an increase in entropy. This complementary nature explains the advantage in combining these two methods (cf. Table 2).

We next present ablation studies on weight decay regularization in Table 3, showing that the proposed approaches improve churn on models trained both with and without weight decay.

**Fixed initialization.** Earlier in Table 1, we showed that fixing the initialization of five runs of the baseline model lowers churn from $5.81 \pm 0.08$ to $5.56 \pm 0.12$ for CIFAR-10 on ResNet-56. However, our proposed Co-distill$_{\mathrm{SKL}}$ model is more effective, reducing churn to $4.29 \pm 0.14$ (see Table 2). Using fixed initialization on this model does not significantly affect churn ($4.24 \pm 0.07$).

## 5 DISCUSSION

**Connection to label smoothing and calibration.** Max entropy regularizers, such as *label smoothing*, are often used to reduce prediction confidence (Pereyra et al., 2017; Müller et al., 2019) and provide better prediction calibration. The minimum entropy regularizers studied in this paper have the opposite effect and increase the prediction confidence, whereas co-distillation reduces the prediction confidences. To study this more concretely we measure the effect of churn reduction methods on the calibration.

We compute Expected Calibration Error (ECE) (Guo et al., 2017) for methods considered in this paper and report the results in Table 4. We notice that the minimum entropy regularizers, predictably, increase the calibration error, whereas the propose co-distillation approach (Co-distill$_{\mathrm{SKL}}$) significantly reduces the calibration error. We notice that our joint approach is competitive with the ensemble distillation approach on CIFAR datasets but incurs higher error on ImageNet. Developing approaches that jointly optimize for churn and calibration is an interesting direction of future work.

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

| Dataset | Method | $\alpha$ | $\beta$ | Temperature | Accuracy | Churn(%) |
|---|---|---|---|---|---|---|
| CIFAR-10 ResNet-56 | Baseline | - | - | - | 93.97$\pm$0.11 | 5.72$\pm$0.18 |
| | Entropy | 0.15 | - | - | 94.05$\pm$0.18 | 5.59$\pm$0.21 |
| | SKL | 0.05 | - | - | 93.75$\pm$0.13 | 5.79$\pm$0.18 |
| | 2-Ensemble distillation | - | - | 3.0 | 94.62$\pm$0.07 | 4.47$\pm$0.13 |
| | Co-distill$_{CE}$ | - | 0.01 | - | 94.25$\pm$0.15 | 5.14$\pm$0.14 |
| | Co-distill$_{SKL}$ | - | 0.01 | - | 94.63$\pm$0.15 | 4.29$\pm$0.14 |
| | +Entropy | 0.1 | 0.01 | - | **94.63$\pm$0.15** | **4.21$\pm$0.15** |
| CIFAR-100 ResNet-56 | Baseline | - | - | - | 73.26$\pm$0.22 | 26.77$\pm$0.26 |
| | Entropy | 0.25 | - | - | 73.24$\pm$0.29 | 26.55$\pm$0.26 |
| | SKL | 0.35 | - | - | 73.35$\pm$0.3 | 25.62$\pm$0.36 |
| | 2-Ensemble distillation | - | - | 4.0 | 76.27$\pm$0.25 | 18.86$\pm$0.26 |
| | Co-distill$_{CE}$ | - | 0.04 | - | 75.11$\pm$0.17 | 19.55$\pm$0.27 |
| | Co-distill$_{SKL}$ | - | 0.04 | - | **76.58$\pm$0.16** | 17.70$\pm$0.33 |
| | +Entropy | 0.2 | 0.04 | - | 76.53$\pm$0.26 | **17.09$\pm$0.3** |
| ImageNet ResNet-v2-50 | Baseline | - | - | - | 76$\pm$0.11 | 15.22$\pm$0.16 |
| | Entropy | 0.2 | - | - | 76.39$\pm$0.11 | 14.92$\pm$0.09 |
| | SKL | 0.05 | - | - | 75.98$\pm$0.08 | 15.32$\pm$0.11 |
| | 2-Ensemble distillation | - | - | 1.0 | 75.83$\pm$0.08 | 12.92$\pm$0.14 |
| | Co-distill$_{CE}$ | - | 0.02 | - | 76.12$\pm$0.08 | 11.62$\pm$0.21 |
| | Co-distill$_{SKL}$ | - | 0.06 | - | **76.74$\pm$0.06** | 11.45$\pm$0.13 |
| | +Entropy | 0.2 | 0.06 | - | 76.56$\pm$0.02 | **11.05$\pm$0.36** |
| iNaturalist ResNet-v2-50 | Baseline | - | - | - | 60.97$\pm$0.26 | 26.99$\pm$0.25 |
| | Entropy | 0.4 | - | - | **62.74$\pm$0.15** | 25.84$\pm$0.2 |
| | Co-distill$_{CE}$ | - | 0.02 | - | 61.1$\pm$0.11 | 26.89$\pm$0.22 |
| | Co-distill$_{SKL}$ | - | 0.05 | - | 61.59$\pm$0.28 | **21.73$\pm$0.26** |
| SVHN LeNet5 | Baseline | - | - | - | 87.16$\pm$0.43 | 14.3$\pm$0.4 |
| | Entropy | 0.2 | - | - | 87.24$\pm$0.61 | 14.12$\pm$0.42 |
| | Co-distill$_{CE}$ | - | 0.02 | - | 88.93$\pm$0.39 | 9.42$\pm$0.33 |
| | Co-distill$_{SKL}$ | - | 0.02 | - | **89.61$\pm$0.25** | **8.49$\pm$0.25** |

Table 5: In this table we list the hyper-parameter values corresponding to the settings for the results in Table 2. For each method we experiment with a range of hyper-parameters, and 10 independent runs, as described in the Section A and report the results for the setting with the best performance.

## A  EXPERIMENTAL SETUP

**Architecture:** For our experiments we use the standard image classification datasets CIFAR and ImageNet. For our experiments with CIFAR, we use ResNet-32 and ResNet-56 architectures, and for ImageNet we use ResNet-v2-50. These architectures have the following configuration in terms of $(n_{\text{layer}}, n_{\text{filter}}, \text{stride})$, for each ResNet block:

- ResNet-32- [(5, 16, 1), (5, 32, 2), (5, 64, 2)]
- ResNet-56- [(9, 16, 1), (9, 32, 2), (9, 64, 2)]
- ResNet-v2-50- [(3, 64, 1), (4, 128, 2), (6, 256, 2), (3, 512, 2)],

where 'stride' refers to the stride of the first convolution filter within each block. For ResNet-v2-50, the final layer of each block has $4 * n_{\text{filter}}$ filters. We use L2 weight decay of strength $1e$-4 for all experiments.

**Learning rate and Batch size:** For our experiments with CIFAR, we use SGD optimizer with a 0.9 Nesterov momentum. We use a linear learning rate warmup for first 15 epochs, with a peak learning rate of 1.0. We use a stepwise decay schedule, that decays learning rate by a factor of 10, at epoch numbers 200, 300 and 400. We train the models for a total of 450 epochs. We use a batch size of 1024.

For the ImageNet experiments, we use SGD optimizer with a 0.9 momentum. We use a linear learning rate warmup for first 5 epochs, with a peak learning rate of 0.8. We again use a stepwise decay schedule, that decays learning rate by a factor of 10, at epoch numbers 30, 60 and 80. We train the models for a total of 90 epochs. We use a batch size of 1024.

**Data augmentation:** We use data augmentation with random cropping and flipping.

**Parameter ranges:** For our experiments with the min entropy regularizers, we use $\alpha \in [0.05, 0.1, 0.15, 0.2, 0.25, 0.3, 0.35, 0.4, 0.5]$, and report the $\alpha$ corresponding to best churn in Table 2. For our experiments with the co-distillation approach, we use $\beta \in [0.01, 0.02, 0.03, 0.04, 0.05, 0.06, 0.07, 0.08]$, and report the $\beta$ corresponding to best churn in Table 2.

For our experiments on entropy regularized co-distillation, we use a linear warmup of the regularizer coefficient. We use the same range for $\alpha$ as described above, and use only the best $\beta$ value from the earlier experiments.

Finally, we run our experiments using TPUv3. Experiments on CIFAR datasets finish in an hour, experiments on ImageNet take around 6-8 hours.

**Hyper parameters for Table 2:** Finally we list in Table 5 the best hyperparameters used to obtain the results in Table 2. For reference we also include the accuracy and the churn again for all methods.

# B  PROOFS

***Proof of Lemma 1***. Recall from Definition 1 that

$$
\begin{aligned}
\mathrm{Churn}(w_1, w_2) &= \mathbb{P}_{\mathsf{X}}\big[\hat{\mathsf{Y}}_{\mathsf{X},w_1} \neq \hat{\mathsf{Y}}_{\mathsf{X},w_2}\big] = \mathbb{P}_{\mathsf{X},\mathsf{Y}}\big[\hat{\mathsf{Y}}_{\mathsf{X},w_1} \neq \hat{\mathsf{Y}}_{\mathsf{X},w_2}\big] \\
&= \mathbb{P}_{\mathsf{X},\mathsf{Y}}\big[\{\hat{\mathsf{Y}}_{\mathsf{X},w_1} = \mathsf{Y}, \hat{\mathsf{Y}}_{\mathsf{X},w_2} \neq \mathsf{Y}\} \cup \{\hat{\mathsf{Y}}_{\mathsf{X},w_1} \neq \mathsf{Y}, \hat{\mathsf{Y}}_{\mathsf{X},w_1} \neq \hat{\mathsf{Y}}_{\mathsf{X},w_2}\}\big] \\
&\overset{(i)}{\leqslant} \mathbb{P}_{\mathsf{X},\mathsf{Y}}\big[\hat{\mathsf{Y}}_{\mathsf{X},w_1} = \mathsf{Y}, \hat{\mathsf{Y}}_{\mathsf{X},w_2} \neq \mathsf{Y}\big] + \mathbb{P}\big[\hat{\mathsf{Y}}_{\mathsf{X},w_1} \neq \mathsf{Y}, \hat{\mathsf{Y}}_{\mathsf{X},w_1} \neq \hat{\mathsf{Y}}_{\mathsf{X},w_2}\big] \\
&\overset{(ii)}{\leqslant} \mathbb{P}\big[\hat{\mathsf{Y}}_{\mathsf{X},w_2} \neq \mathsf{Y}\big] + \mathbb{P}\big[\hat{\mathsf{Y}}_{\mathsf{X},w_1} \neq \mathsf{Y}\big] \\
&= \mathrm{P}_{\mathrm{Err},\mathsf{w}_2} + \mathrm{P}_{\mathrm{Err},\mathsf{w}_1},
\end{aligned} \tag{10}
$$

where $(i)$ and $(ii)$ follow from the union bound and the fact that $\mathbb{P}[A] \leqslant \mathbb{P}[B]$ whenever $A \subseteq B$, respectively. □

***Proof of Lemma 2***. Note that, for any $j \neq \hat{y}_{x,w_1}$,

$$
\begin{aligned}
f(x;w_2)_{\hat{y}_{x,w_1}} &- f(x;w_2)_j \\
&= f(x;w_2)_{\hat{y}_{x,w_1}} - f(x;w_1)_{\hat{y}_{x,w_1}} + \underbrace{f(x;w_1)_{\hat{y}_{x,w_1}} - f(x;w_1)_j}_{\geqslant \gamma_{x,w_1}} + f(x;w_1)_j - f(x;w_2)_j \\
&\geqslant \gamma_{x,w_1} - \sum_{j \in \mathcal{Y}} |f(x;w_1)_j - f(x;w_2)_j| \\
&= \gamma_{x,w_1} - D_{\mathrm{L1}}\big(f(x;w_1), f(x;w_2)\big).
\end{aligned} \tag{11}
$$

Similarly, for any $j \neq \hat{y}_{x,w_2}$, we can establish that

$$
f(x;w_1)_{\hat{y}_{x,w_2}} - f(x;w_1)_j \geqslant \gamma_{x,w_2} - D_{\mathrm{L1}}\big(f(x;w_1), f(x;w_2)\big). \tag{12}
$$

Note that experiencing churn between the two models on $x$ is equivalent to

$$
\begin{aligned}
\{\hat{y}_{x,w_1} \neq \hat{y}_{x,w_2}\} &\subseteq \\
&\big\{\exists j \neq \hat{y}_{x,w_1} : f(x;w_2)_{\hat{y}_{x,w_1}} < f(x;w_2)_j\big\} \bigcup \big\{\exists j \neq \hat{y}_{x,w_2} : f(x;w_1)_{\hat{y}_{x,w_2}} < f(x;w_1)_j\big\} \\
&\overset{(i)}{\subseteq} \big\{D_{\mathrm{L1}}\big(f(x;w_1), f(x;w_2)\big) > \gamma_{x,w_1}\big\} \bigcup \big\{D_{\mathrm{L1}}\big(f(x;w_1), f(x;w_2)\big) > \gamma_{x,w_2}\big\},
\end{aligned} \tag{13}
$$

where $(i)$ follows from (11) and (12). Now, (13) implies that

$$
\mathbb{P}_{\mathsf{X}}\{\hat{\mathsf{Y}}_{\mathsf{X},w_1} \neq \hat{\mathsf{Y}}_{\mathsf{X},w_2}\} \leqslant \mathbb{P}_{\mathsf{X}}\big[D_{\mathrm{L1}}(f(\mathsf{X};w_1), f(\mathsf{X};w_2)) > \min\{\gamma_{\mathsf{X},w_1}, \gamma_{\mathsf{X},w_2}\}\big]. \tag{14}
$$
□

***Proof of Theorem 3***. Let the prediction for a given $x$ be $p = f(x;w)$. W.L.O.G. let $p \geqslant 1 - p$. The prediction confidence is then

$$
\gamma_{x,w} = p - (1 - p) = 2p - 1. \tag{15}
$$

**Entropy case.** Now we can write entropy in terms of the prediction confidence (cf. (15))

$$
\begin{aligned}
H(f(x;w)) &= -p\log(p) - (1-p)\log(1-p) \\
&= -\frac{(1+\gamma_{x,w})}{2} \times \log\left(\frac{1+\gamma_{x,w}}{2}\right) - \frac{(1-\gamma_{x,w})}{2} \times \log\left(\frac{1-\gamma_{x,w}}{2}\right) \\
&\triangleq g(\gamma_{x,w}).
\end{aligned}
\tag{16}
$$

Now the gradient of $g(\cdot)$ is $\nabla_{\gamma_{x,w}} g(\gamma_{x,w}) = \frac{1}{2}\log\left(\frac{1-\gamma_{x,w}}{1+\gamma_{x,w}}\right)$, which is less than 0 for $\gamma_{x,w} \in [0,1]$. Hence, the function $g(\gamma_{x,w})$ is a decreasing function for inputs in range $[0,1]$. Hence, if $g(\gamma_{x,w'}) \geqslant g(\gamma_{x,w})$ implies $\gamma_{x,w'} \leqslant \gamma_{x,w}$.

**Symmetric-KL divergence case.** Recall that

$$
\begin{aligned}
\mathrm{SKL}(f(x;w), \mathrm{Unif}) &= \mathrm{KL}\big(f(x;w) \| \mathrm{Unif}\big) + \mathrm{KL}\big(\mathrm{Unif} \| f(x;w)\big) \\
&= \sum\nolimits_{j \in [K]} \big(f(x;w)_j - 1/K\big) \cdot \log\big(K \cdot f(x;w)_j\big) \\
&= \sum_j f(x;w)_j \cdot \log\big(f(x;w)_j\big) - \frac{1}{K}\log\big(f(x;w)_j\big).
\end{aligned}
$$

For binary classification this reduces to,

$$
\mathrm{SKL}(f(x;w), \mathrm{Unif}) = -\frac{1}{2}\log(p(1-p)) - H(p).
$$

By using (15) and (16), we can rewrite this function in terms of $\gamma_{x,w}$ as:

$$
\begin{aligned}
\mathrm{SKL}(f(x;w), \mathrm{Unif}) &= -\frac{1}{2}\log(p(1-p)) - H(p) \\
&= -\frac{1}{2}\cdot\log\left(\frac{1+\gamma_{x,w}}{2} \times \frac{1-\gamma_{x,w}}{2}\right) - g(\gamma_{x,w}) = -\frac{1}{2}\cdot\log\left(\frac{1-\gamma_{x,w}^2}{4}\right) - g(\gamma_{x,w}).
\end{aligned}
$$

Now notice that both the above terms are an increasing function of $\gamma_{x,w}$, as log is an increasing function, and $g(\gamma_{x,w})$ is a decreasing function. Hence $\mathrm{SKL}(f(x;w), \mathrm{Unif}) \geqslant \mathrm{SKL}(f(x;w'), \mathrm{Unif})$ implies $\gamma_{x;w} \geqslant \gamma_{x;w'}$. $\qquad\square$

## C   DESIGN CHOICES IN THE PROPOSED VARIANT OF CO-DISTILLATION

Here, we discuss two important design choices that are crucial for successful utilization of co-distillation framework for churn reduction while also improving model accuracy.

**Joint vs. independent updates.** Anil et al. (2018) consider co-distillation with independent updates for the participating models. In particular, with two participating models, this corresponds to independently solving the following two sub-problems during the $t$-th step.

$$
w_1^t = \arg\min\nolimits_{w_1 \in \mathcal{W}} L(w_1; \mathcal{S}) + \beta \cdot \frac{1}{n}\sum\nolimits_{i \in [n]} \mathrm{KL}\big(f(x_i; w_2^{t-T}) \| f(x_i; w_1)\big)
\tag{17a}
$$

$$
w_2^t = \arg\min\nolimits_{w_2 \in \mathcal{W}} L(w_2; \mathcal{S}) + \beta \cdot \frac{1}{n}\sum\nolimits_{i \in [n]} \mathrm{KL}\big(f(x_i; w_1^{t-T}) \| f(x_i; w_2)\big),
\tag{17b}
$$

where $w_1^{t-T}$ and $w_2^{t-T}$ corresponds to earlier checkpoints of two models being used to compute the model disagreement loss component at the other model. Note that since $w_2^{t-T}$ and $w_1^{t-T}$ are not being optimized in (17a) and (17b), respectively. Thus, for each model, these objectives are equivalent to regularizing its empirical loss via a cross-entropy terms that aims to align its prediction probabilities with that of the other model. In particular, the optimization steps in (17) are equivalent to

$$
w_1^t = \arg\min\nolimits_{w_1 \in \mathcal{W}} L(w_1; \mathcal{S}) + \beta \cdot \frac{1}{n}\sum\nolimits_{i \in [n]} \mathrm{H}\big(f(x_i; w_2^{t-T}), f(x_i; w_1)\big)
\tag{18a}
$$

$$
w_2^t = \arg\min\nolimits_{w_2 \in \mathcal{W}} L(w_2; \mathcal{S}) + \beta \cdot \frac{1}{n}\sum\nolimits_{i \in [n]} \mathrm{H}\big(f(x_i; w_1^{t-T}), f(x_i; w_2)\big),
\tag{18b}
$$

where $H\big(f(x; w_1), f(x; w_2)\big) = -\sum_j f(x; w_1)_j \log f(x; w_2)_j$ denotes the cross-entropy between the probability mass functions $f(x; w_1)$ and $f(x; w_2)$.

In our experiments, we verify that this independent updates approach leads to worse results as compared to *jointly* training the two models using the objective in (9).

**Deferred model disagreement loss component in co-distillation.** In general, implementing co-distillation approach by employing the model disagreement based loss component (e.g., $\mathrm{SKL}\big(f(x; w_1), f(x; w_2)\big)$ in our proposal) from the beginning leads to sub-optimal test accuracy as the models are very poor classifiers at the initialization. As a remedy, Anil et al. (2018) proposed a step-wise ramp up of such loss component, i.e., a time-dependent $\beta(t)$ such that $\beta(t) > 0$ iff $t > t_0$, with $t_0$ burn-in steps. Besides step-wise ramp up, we experimented with linear ramp-up, i.e., $\beta_t = \min\{c \cdot t, \beta\}$ and observed that linear ramp up leads to better churn reduction. We believe that, for churn reduction, it's important to ensure that the prediction probabilities of the two models start aligning before the model diverges too far during the training process.

## D  COMBINED APPROACH: CO-DISTILLATION WITH ENTROPY REGULARIZER

In this paper, we have considered two different approaches to reduce churn, involving minimum entropy regularizers and co-distillation framework, respectively. This raises an important question if these two approaches are redundant. As shown in Figures 2, 1, and 3, this is not the case. In fact, these two approaches are complementary in nature and combining them together leads to better results in term of both churn and accuracy.

Note that the minimum entropy regularizers themselves cannot *actively* enforce alignment between the prediction probabilities across multiple runs as the resulting objective does not couple multiple models together. On the other hand, as verified in Figure 2, the co-distillation framework in itself does not affect the prediction confidence of the underlying models. Thus, both these approaches can be combined together to obtain the following training objective which promotes both large model prediction confidence and small variance in prediction probabilities across multiple runs.

$$L_{\mathrm{Co\text{-}distill}_{\mathrm{SKL}}}^{\mathrm{combined}}(w_1, w_2; \mathcal{S}) = L_{\mathrm{Co\text{-}distill}_{\mathrm{SKL}}}(w_1, w_2; \mathcal{S})(w_1, w_2; \mathcal{S}) + \alpha \cdot \mathrm{EReg}(w_1, w_2; \mathcal{S}), \quad (19)$$

where

$$\mathrm{EReg}(w_1, w_2; \mathcal{S}) = \frac{1}{n} \sum\nolimits_{i \in [n]} H\big(f(x_i, w_1)\big) + H\big(f(x_i, w_2)\big)$$

or

$$\mathrm{EReg}(w_1, w_2; \mathcal{S}) = -\frac{1}{n} \sum\nolimits_{i \in [n]} \Big(\mathrm{SKL}\big(f(x_i; w_1), \mathrm{Unif}\big) + \mathrm{SKL}\big(f(x_i; w_2), \mathrm{Unif}\big)\Big).$$

## E  ADDITIONAL EXPERIMENTAL RESULTS

In this section we present additional experimental results.

**Ablation on CIFAR-100.** Next we present our ablation studies, similar to results in Figure 1, for CIFAR-100. In Figure 3, we plot the mean prediction entropy, accuracy and churn, for the proposed SKL regularizer (7) and the co-distillation approach (9), for varying $\alpha$ and $\beta$, respectively. Similar to Figure 1, the plots show the effectiveness of the proposed approaches in reducing churn, while improving accuracy.

### E.1  ON HARDWARE NONDETERMINISM: GPU BASELINE ABLATION RESULTS

We now repeat the experiments in Sec. 2.3 on GPUs to investigate the unavoidable nondeterminism introduced by different hardware platforms.

We report results in Table 6. Compared to Table 1, there are two main differences. First, these experiments are run on GPU with smaller ResNet-32 models. Second, these results always use **data**

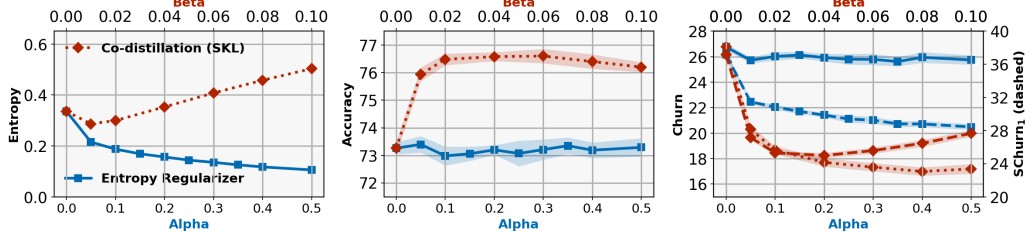

Figure 3: CIFAR-100 ablation study: Similar to Figure 1, we plot the effect of the SKL regularizer (cf. (7)), for varying $\alpha$, and our proposed variant of co-distillation (Co-distill$_{SKL}$) for varying $\beta$, on the prediction entropy, accuracy, and churn. These plots shows the *complementary* nature of these methods in reducing churn. While the regularizer reduces churn, by reducing the prediction entropy, Co-distill$_{SKL}$ reduces churn by improving the agreement between two models, hence increasing the entropy.

| | | | | |
|---|---|---|---|---|
| **Identical initialization** | | | ✓ | ✓ |
| **Ident. input data** | | ✓ | | ✓ |
| Accuracy | $91.67 \pm 0.22$ | $91.61 \pm 0.16$ | $91.53 \pm 0.13$ | $91.86 \pm 0.05$ |
| Churn(%) | $8.44 \pm 0.27$ | $8.25 \pm 0.11$ | $2.73 \pm 0.07$ | $1.33 \pm 0.10$ |
| SChurn$_1$(%) | $8.71 \pm 0.26$ | $8.69 \pm 0.12$ | $2.98 \pm 0.08$ | $1.47 \pm 0.05$ |

Table 6: Ablation study of churn across 5 runs on CIFAR-10 with a ResNet-32 on GPU. Holding the initialization constant across models decreases churn significantly, and using identical input data (keeping minibatch ordering and augmentation constant) lowers churn further. Unlike Table 1, these experiments were performed on GPU rather than TPU. Under this setting, using fixed initialization does achieve a lower churn at a much more expensive computation cost.

**augmentation**, but control for the randomness differently by ensuring that all models within each run perform the same augmentations during training; we call these "identical input data" ablations because they control for both minibatch ordering and data augmentation. Again, the the first column gives the baseline churn with no factor held constant across runs, and the last column has every possible source of churn except hardware nondeterminism held constant across runs.

**Model initialization** is a significant source of churn. Simply initializing weights from identical seeds (columns 1-2 in Table 1) can significantly decrease churn, no matter what other aspects of training are held constant. Holding input data constant across runs does reduce randomness and further reduces churn, but is less significant compared to constant initialization.

The rightmost column eliminates all possible sources of churn except for unavoidable non-determinism in the computing platform used for training machine learning models, e.g., GPU/TPU (Morin & Willetts, 2020; PyTorch, 2019; Nvidia, 2020). The experiments in Table 6 were run on GPU, and even when all other aspects of training are held constant (rightmost column), model weights diverge within 100 steps (across runs) and the final churn is significant. We verified that this is the sole source of non-determinism: models trained to 10,000 steps on CPUs under these settings had identical weights. Comparing these results to the rightmost column of Table 1, it seems TPU platforms currently introduce a higher baseline amount of churn than GPUs. However, as shown in this work, our Co-distill$_{SKL}$ method can alleviate this somewhat.

### E.2  2 Ensemble

In Table 7 we provide the results for 2-ensemble models and compare with distillation and co-distillation approaches. We notice that the proposed approach with entropy regularizer and co-distillation achieve similar or better churn, with lower inference costs. However the the ensemble models achieve better accuracy on certain datasets, and could be an alternative in settings where higher inference costs are acceptable.

| Dataset | Method | Training cost | Inference cost | Accuracy | Churn(%) | SChurn$_1$(%) | Churn correct | Churn incorrect |
|---|---|---|---|---|---|---|---|---|
| CIFAR-10 ResNet-56 | Baseline | 1x | 1x | 93.97±0.11 | 5.72±0.18 | 6.41±0.15 | 2.62±0.12 | 53.82±1.44 |
| | 2-Ensemble | 2x | 2x | **94.79±0.09** | **4.02±0.12** | 5.42±0.09 | **1.84±0.11** | 43.74±1.24 |
| | 2-Ensemble distil. (Hinton et al., 2015) | 3x | 1x | 94.62±0.07 | 4.47±0.13 | 4.91±0.11 | 2.05±0.1 | 47.25±1.55 |
| | Co-distill$_{CE}$ (Anil et al., 2018) | 2x | 1x | 94.25±0.15 | 5.14±0.14 | 5.62±0.12 | 3.03±0.31 | 40.45±5.98 |
| | Co-distill$_{SKL}$ (this paper) | 2x | 1x | 94.63±0.15 | 4.29±0.14 | 4.66±0.11 | 2.49±0.25 | **36.78±4.14** |
| | +Entropy (this paper) | 2x | 1x | 94.63±0.15 | 4.21±0.15 | **4.61±0.14** | 1.94±0.13 | 44.29±1.9 |
| CIFAR-100 ResNet-56 | Baseline | 1x | 1x | 73.26±0.22 | 26.77±0.26 | 37.19±0.35 | 11.42±0.26 | 68.77±0.72 |
| | 2-Ensemble | 2x | 2x | 76.28±0.24 | 20.23±0.29 | 35.91±0.33 | 8.26±0.25 | 58.92±0.83 |
| | 2-Ensemble distil. (Hinton et al., 2015) | 3x | 1x | 76.27±0.25 | 18.86±0.26 | 25.43±0.23 | 7.79±0.23 | 54.3±0.86 |
| | Co-distill$_{CE}$ (Anil et al., 2018) | 2x | 1x | 75.11±0.17 | 19.55±0.27 | 25.03±0.27 | 9.98±0.97 | 48.69±2.84 |
| | Co-distill$_{SKL}$ (this paper) | 2x | 1x | **76.58±0.16** | 17.70±0.33 | 24.99±0.31 | 9.08±0.8 | **45.98±2.98** |
| | +Entropy (this paper) | 2x | 1x | 76.53±0.26 | **17.09±0.3** | 24.06±0.28 | **7.2±0.26** | 49.4±1.1 |
| ImageNet ResNet-v2-50 | Baseline | 1x | 1x | 76±0.11 | 15.22±0.16 | 35.01±0.24 | 5.87±0.11 | 44.77±0.46 |
| | 2-Ensemble | 2x | 2x | **77.24±0.09** | 11.13±0.13 | 27.71±0.18 | **4.2±0.11** | 34.63±0.43 |
| | 2-Ensemble distil. (Hinton et al., 2015) | 3x | 1x | 75.83±0.08 | 12.92±0.14 | 34.02±0.2 | 4.92±0.11 | 38.04±0.39 |
| | Co-distill$_{CE}$ (Anil et al., 2018) | 2x | 1x | 76.12±0.08 | 11.62±0.21 | **18.38±0.34** | 4.97±0.36 | 32.67±1.45 |
| | Co-distill$_{SKL}$ (this paper) | 2x | 1x | 76.74±0.06 | 11.45±0.13 | 28.66±0.20 | 5.11±0.36 | **32.41±1.15** |
| | +Entropy (this paper) | 2x | 1x | 76.56±0.02 | **11.05±0.36** | 22.92±1.01 | 4.32±0.32 | 32.89±1.15 |

Table 7: Similar to Table 2, we provide results on accuracy and churn for 2-ensemble and proposed approaches. We notice that proposed approaches are either competitive or improve churn over 2-ensemble method, while keeping the inference costs low.

## F   LOSS LANDSCAPE: ENTROPY REGULARIZATION VS. TEMPERATURE SCALING

Figure 4 visualises the entropy regularised loss (6) in binary case. We do so both in terms of the underlying loss operating on probabilities (i.e., log-loss), and the loss operating on logits with an implicit transformation via the sigmoid (i.e., the logistic loss). Here, the regularised log-loss is $\ell\colon [0, 1] \to \mathbb{R}_+$, where $\ell(p) = (1 - \alpha) \cdot -\log p + \alpha \cdot H_{\mathrm{bin}}(p)$, for binary entropy $H_{\mathrm{bin}}$. Similarly, the logistic loss is $\ell\colon \mathbb{R} \to \mathbb{R}_+$, where $\ell(f) = (1 - \alpha) \cdot -\log \sigma(f) + \alpha \cdot H_{\mathrm{bin}}(\sigma(f))$, for sigmoid $\sigma(\cdot)$. The effect of increasing $\alpha$ is to dampen the loss for high-confidence predictions – e.g., for the logistic loss, either very strongly positive or negative predictions incur a low loss. This encourages the model to make confident predictions.

Figure 5, by contrast, illustrates the effect of changing the temperature $\tau$ in the softmax. Temperature scaling is a common trick used to decrease classifier confidence. This also dampens the loss for high-confidence predictions. However, with strongly negative predictions, one still incurs a high loss compared to strongly positive predictions. This is in contrast to the more aggressive dampening of the loss as achieved by the entropy regularised loss.

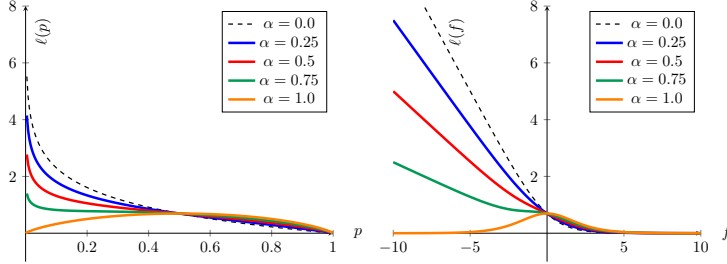

Figure 4: Visualisation of entropy regularised loss (eq. (6)) in binary case. On the left panel is the regularised log-loss $(1 - \alpha) \cdot -\log p + \alpha \cdot H_{\mathrm{bin}}(p)$, which accepts a probability in $[0, 1]$ as input. On the right panel is the logistic loss $(1 - \alpha) \cdot -\log \sigma(f) + \alpha \cdot H_{\mathrm{bin}}(\sigma(f))$, which accepts a score in $\mathbb{R}$ as input and passes it through sigmoid $\sigma(\cdot)$ to get a probability estimate.

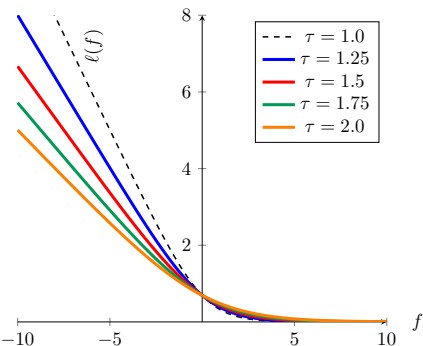

Figure 5: Visualisation of temperature scaling on binary logistic lau. We plot $-\log \sigma(f/\tau)$ for varying temperature parameter $\tau$.

