# OpenReview forum: "On the Reproducibility of Neural Network Predictions"
_ICLR.cc/2021/Conference — Reject_

### Official Review · AnonReviewer3 · 2020-10-28
**How does correcting for churn affect calibration of probability estimates?**

**Rating:** 4
**Confidence:** 3

**Review:**

The paper proposes methods to address churn in deep neural networks for classification, defined as the extent of disagreements in predictions of two models trained on the same data with the same algorithm. In addition to an existing measure of churn that is based on exact match of predicted classes, the paper introduces a soft measure of churn that measures disagreement by comparing the two models' class probability distributions. The paper proposes three regularization terms that can be added to the primary loss function used during training to reduce churn: two single-model regularization terms (based on cross entropy and KL divergence respectively) that encourage the model to output a more uneven probability distribution for an example, and a divergence-based term that is used by training two models simultaneously and that imposes a KL-divergence-based penalty that encourages the two models to output probability distributions that are as similar as possible. Experiments with ResNet architectures on CIFAR-10/100 and ImageNet indicate that the proposed approaches and their combination indeed reduce churn and do so to a larger extent than the two-model divergence-based approach applied in conjunction with cross-entropy that was proposed in work by Anil et al. in 2018 (although the improvement seems quite minor on ImageNet).

My primary concern is that the paper does not evaluate the effect of the regularization terms on calibration of the probability estimates. This issues is briefly discussed in Section 5. In practice, it seems vastly more important to have well-calibrated probability estimates than no churn. The single-model regularization terms proposed in the paper seem to encourage the model to become overconfident. This will reduce churn but is clearly not desirable in most practical applications.

Another concern is that the hyperparameters \alpha and \beta appear to be tuned by maximizing performance on the *test* data. This inflates the performance estimates on this data obtained for the proposed methods compared to the baseline, which does not have these hyperparameters to play with.

I would also like to see results for churn when standard forms of regularization are included, such as L_2 or L_1 regularization. As these should also increase the stability of the learning process, it stands to reason that they will also reduce churn (particularly when the hyperparameters are tuned on the test data).

The paper appears to claim that the ordering of the mini batches (over multiple runs!) has an effect on accuracy. This does not appear to make sense (assuming only random orders are considered).

The paper eliminates churn due to data augmentation by removing data augmentation. Instead, the random number generator used for augmentation should be initialized in a deterministic manner.

Given that the best-performing model is based on training two networks, the paper should also include results for two-member ensembles that are obtained by averaging the probability distributions across the two networks. If memory consumption and inference time are not critical, and this two-member ensemble turns out to reduce churn substantially, it would be a useful solution.

Some small issues:

"disagreements between predictions of the two models independently trained by the same algorithm" --- on the same training data???

Table 2 is discussed before Table 1.

Why is SChurn_1 denoted a percentage in Table 1?

---

> ### Author Response · Authors · 2020-11-17
> **Thanks for the comments**
>
> We thank the reviewer and have updated the draft based on their comments and suggestions.
>
> 1. **Calibration**. Following reviewers suggestion we computed the Expected Calibration Error for different methods considered in the paper  and presented the results in Table 4 (Section 5). We note that while the minimum entropy regularizers, predictably, increase the calibration error, our combined approach with co-distillation results in calibration error competitive with 2-ensemble distillation method. We emphasize that having both lower churn and well calibrated estimates are important properties of a ML model. Calibrated predictions that are sensitive to random seed of training algorithms may make them less useful in applications. Hence, having methods that both reduce churn and improve calibration are of interest. We plan to pursue a detailed study on this interaction in future works.
>
> 2. **Hyperparameters**. We note that the learning rate, momentum, learning rate schedule, weight decay and the temperature for distillation parameters were chosen based on best performance of the baseline. For example, our baseline accuracy for Imagenet using Resnet50 matches the results from [1]. We **do not tune** these parameters for our experiments with the proposed methods. Given, we chose all the above key parameters based on best performance of the baseline, we believe this is a fair comparison. We experiment with alpha in range [0.05, 0.5] with 0.05 increments and beta in range [0.01, 0.08] with 0.01 increments. These details are given in Section A in appendix.
>
> 3. **Regularization**. We note that all models in the paper are trained with standard regularization techniques such as dropout and L_2 regularization (weight decay) (Ref. Section A in appendix).  On the reviewer’s suggestion, we added an ablation study and included the results for models trained *without* weight decay in Table 3 (Section 4). We notice that the proposed techniques improve churn both with and without weight decay, showing the effect of the proposed techniques is orthogonal to weight decay.
>
> 4. **Minibatch ordering and Data augmentation**. We agree with the reviewer and indeed have included experiments that hold the minibatch order and data augmentation the same across runs, in Section E.1 of the submission. These experiments however are done on GPU and not on TPUs, which were used for the experiments in the remainder of the paper.  By keeping the same data order and data augmentation across runs, we are able to preserve the accuracy and reduce churn (from 8.5% to 1.3%). Due to some technical challenges we have been delayed in doing similar experiments on TPU and plan to include them in the final version. Please also see the comment to Reviewer 1.
>
> 5. **Two member ensemble**. We have added a table (#7) in the appendix (Section E) with the results from the two member ensembles. We notice that proposed approaches are either competitive or improve churn over the 2-ensemble method. However, the 2-ensemble models have better accuracy on some datasets at the cost of expensive inference.
>
> [1] https://pytorch.org/docs/stable/torchvision/models.html

---

### Official Review · AnonReviewer4 · 2020-10-31
**interesting and useful findings but not comprehensive enough empirical validation**

**Rating:** 5
**Confidence:** 2

**Review:**

The paper investigates two methods to reduce churn in neural network classification prediction. Churn is when two networks trained on the same data produce outputs that disagree, due to randomness in the training process. The authors identify several sources of randomness, from underlying hardware differences to parameter initialization and more. The authors propose two ways to mitigate churn. One is to use entropy minimization to favor more confident predictions. The second is to use co-distillation, a form of online ensemble learning. The authors show that both together do a good job of reducing churn on three data sets.

The paper does a decent job of making an important point about churn, investigating its prevalence, and proposing a solution. The approach is sound and promising. For a narrow result like this, specific to one metric of neural networks, I would like much more empirical validation that the authors provide. Only three data sets and three baselines does not seem like enough, given that the experiments provide the main take-home message of the paper.

I'd like a deeper discussion about why churn is bad. Can the authors give a concrete example where churn will make a machine learning system more undesirable? For example, imagine a facial recognition system. What if new data or new training lead to a new model that is just as accurate but makes mistakes on different people than before. In what application is that inherently bad? Can you formulate the problem with churn more formally? In the current paper, it's mostly assumed to be undesirable. To an extent, I agree, but I'd like to understand more clearly why it is undesirable. I think the comparison to reproducible scientific experiments is a little loose. A machine learning algorithm is not a scientific experiment. I don't think the authors need to cite quite so many papers about the much more general problem of reproducibility in science.

This finding is interesting and instructive: "churn observed in Table 2 is not merely caused by the discontinuity of the arg max".

This finding is fascinating: "Even with extreme measures to eliminate all sources of randomness, we continue to observe churn due to unavoidable hardware non-determinism."

I have a question about the minimum entropy procedure. Doesn't it depend on the confidence scores being accurate? For example, if some scores were overconfident, the minimum entropy procedure would tend to select those predictions and reduce accuracy. Imagine that confidence scores are normally distributed: some confidence scores are accurate but some are under- or over-confident. Minimum entropy will tend to pick the over-confident ones even though the confidence is in error. Is this a real danger and do the authors observe this at all with their technique?

This is more a question for Cormier et al. than for the current authors, but why use the word "churn" instead of "disagreement"? Is there a difference? From what I can tell, churn and disagreement are the same thing, and churn has a different English meaning. Disagreement seems like the better term for this.

The following paper explored the use of disagreement as a model selection tool and I think may have also proven Lemma 1:

https://papers.nips.cc/paper/2603-co-validation-using-model-disagreement-on-unlabeled-data-to-validate-classification-algorithms

Minor comments and typos:

Why is Table 2 on page 7 when it is referred to on page 3? Why is Table 2 referred to before Table 1?

"linear warmup and join updates"
Did you mean?:
linear warmup and joint updates

"any of the participating model can be used for inference"
Did you mean?:
any of the participating models can be used for inference

worst cast bound
worst-case bound

runs.We
runs. We

Intuitively,encouraging
Intuitively, encouraging

---

> ### Author Response · Authors · 2020-11-17
> **Thanks for the comments**
>
> We thank the reviewer and have updated the draft based on their comments and suggestions.
>
> 1. **Additional datasets**. Following reviewers suggestion we include experimental results from two additional datasets SVHN and iNaturalist, in Table 2. We again notice that the proposed methods are effective at reducing churn improving over the existing baselines. Interestingly we notice that the entropy regularizers improve accuracy significantly on iNaturalist, similar to Imagenet, showing their effectiveness beyond reducing churn. All experiments are repeated with 10 independent runs and we report mean and standard deviation across these runs. We note that in addition to these datasets, the paper also includes ablation studies on different aspects of training algorithms, such as initialization, minibatch ordering and data augmentation, to identify the causes of churn.
>
> 2. **Motivation**. In machine learning, beyond reproducibility, it is desirable to not have model predictions sensitive to the random seed used by the training algorithms [1]. In the facial recognition example, consider the user perspective and their experience. Having a system that with every update either recognizes them or not, will lead to a frustrating experience. In this case aggregate accuracy does not capture the experience of such individual users, and it is desirable to have changes that reduce churn over the ones that increase churn. Moreover, having consistent success/failure cases for a model makes it easier to interpret the model predictions and develop techniques to potentially improve it.
>
>
> 3. **Confidence**. We do not fully understand the reviewers comment about certain confidence scores being accurate. Since we are only given 0, 1 labels, we are not sure what it means to have accurate prediction confidences, apart from the standard 0/1 accuracy. The entropy regularizers mainly affect the examples near the classification boundary, and push them away from it. This makes it harder for model’s predictions to flip due to small perturbations during training. If the reviewer’s concerns were about the calibration, we do agree that the entropy regularizer makes the calibration error wose. However combined with co-distillation we notice that the joint procedure has comparable calibration error as the ensemble distillation approach (please see Table 4 and also the response to R3).
>
> 4. **Disagreement**. We agree that alternatively we can call Churn as Disagreement between models. We were following the previous works (Cormier et al. 2016, Anil et al., 2018), in re-using the definition of Churn in this paper.
>
> Thanks for pointing to the reference and corrections. We have updated the paper accordingly.  We added the reference in both the introduction and at Lemma 1.
>
> [1] Peter Henderson, Riashat Islam, Philip Bachman, Joelle Pineau, Doina Precup, and David Meger.Deep reinforcement learning that matters. In Thirty-Second AAAI Conference on Artificial  Intelligence, 2018.

---

### Official Review · AnonReviewer1 · 2020-11-02
**Reasonable breadth of empirical analysis but experimental design leads to potentially misleading results.**

**Rating:** 5
**Confidence:** 4

**Review:**

This work studies the ‘churn’ (disagreement between predictions of two replicates) caused by different sources of variation in the training procedure and proposes solutions to reduce it. One solution is to use minimum entropy regularization to increase prediction confidences and the second solution is to force model agreement via co-distillation.

===============================

Pros:
1. The paper is well written and clear.
2. Studies dissect many components, e.g. churn caused different sources of variation, ablation study of the proposed co-distillation+entropy.
3. Results are compared to reasonable baselines.

Cons:
1. I find the problem statement unconvincing. How much retraining from scratch affects generalization? Beside variability on the test set accuracy, is there any evidence that the fluctuations observed are representative of variations of the true risk (on all data distribution)? Or is it only noise due to the small size of the test set?
2. Modification of two independent variables (dependent and independent variables in planning of experimental procedures) in the experiments of Table 1 likely make the results misleading. (More on this below)

===============================

Reasons for score:

I would vote for a weak reject. The breadth of the empirical analysis is sufficient but subtle details (as further explained below) make the results misleading for Table 1.

===============================

Additional observations

Modification of two independent variables (dependent and independent variables in planning of experimental procedures) in the experiments of Table 1 make the results potentially misleading. The churn depends to some degree on the level of accuracy, and data augmentation significantly affects accuracy as well. Indeed, removing data augmentation leads to a drastic drop of 4% accuracy. In the same way, but less drastic, the random data order from one epoch to another affects accuracy. When modifying data augmentation or data order, it is not possible to determine whether the change of churn is strictly due to data augmentation/data order or to accuracy drop. One way of avoiding this confounding effect would be to conduct hyperparameter optimization in a way to enforce a given level of accuracy (ex: 88%). We could then compare with augmentation at 88% accuracy vs no augmentation at 88% accuracy. If for instance a sub-optimal learning rate with data augmentation yielding 88% accuracy leads to the same level of churn than a good learning rate with no data augmentation yielding 88% accuracy, then we could conclude the effect on churn is mainly due to accuracy itself. I would nevertheless assume data augmentation to reduce churn indeed as it can be seen as increasing the dataset size, which reduces the level of noise. Therefore, to measure the effect of accuracy alone on churn I would also run two experiments where I optimize the learning rate to find accuracies of 84% and 88% (both without data augmentation) to see the relation between accuracy and churn on equal dataset size.

On the same topic, the authors should avoid removing altogether data augmentation when they want to remove its effect of variation. They should rather seed it. I understand it is more effort as I recently went through the process, but it is possible. This way they would study the effect of varying data augmentation vs fixing it across replicates without losing its regularization effect. The same applies for data order. It is possible to seed data order without losing the randomization from one epoch to another.

In section 2.3, the authors say that ‘Even when all other aspects of training are held constant (rightmost column), model weights diverge within 100 steps (across runs) and the final churn is significant.’ I am not familiar with TensorFlow, but I know ResNet implementations based on the cudnn backend require the deterministic operators for perfect reproducibility. ResNet is trainable in a deterministic way using PyTorch for instance. It turns out I have also studied these sources of noise and the residual variance resulting from this noise (numerical noise due to operation order on GPU) is significantly smaller than the one caused by data sampling, weighs init, data ordering or data augmentation. The results last column on table 1, where churn is only caused by numerical noise, suggests that the large increase in churn when removing data augmentation is mainly due to the decrease in accuracy. When only numerical noise is present there is smaller variance in results so I would expect churn to be smaller. The fact that it increases in Table 1 suggests to me that we are indeed observing a confounding effect where the main cause is the drop of accuracy rather that the removal of data augmentation.


Bold results in table 2 are misleading. Many of them are not significantly different yet only one result per column is in bold. All top results that are not significantly different should be in bold.

The co-distillitation + entropy procedure proposed in this paper introduces 2 new hyperparameters. The optimization of these hyper-parameters can lead to misleading results if hyperparemeters of baselines are not optimized accordingly with similar budgets. The experimental section should report these optimization procedures so that we can assert the reliability of the results. Also, were the alpha and beta optimized to provide better accuracy of lower churn in Table 2?

===============================

Typos, minor comments, questions

Intro in section 2 should restate that the definition builds open the work of Cormier et al 2016.

In the co-distillation process, how do you choose which model to retain to compute the churn? As I understand it the churn is computed on 2 models that are trained independently with other co-distillation ‘siblings’. Do you pick randomly within the co-distillation pairs?

Page 1, first paragraph: a a novel -> a novel
Page 5, Intuitively,encouraging -> Intuitively, encouraging

===============================

Post-Rebuttal

I thank the authors for the detailed answer. In light of the response of the authors and the other reviews, I still recommend rejecting the paper, with a rating of 5.

Some comments based on the rebuttal:

I find the data in table 6 to support my point on confounding variables. The churn with data augmentation fixed on GPU is systematically higher than with random data augmentation on TPU when model initialization is random. Note that the main differences here beside the fact that data augmentation is fixed or random, is that the accuracy is lower by 1.5%-2.5%. We see again an increase in churn related to a decrease in accuracy. Just as when the data augmentation was removed altogether. The authors say that accuracy change itself isn't predictive of churn because we could make a training perfectly reproducible with lower accuracy thus leading to 0 churn, but the same argument would hold for removing a fixed data augmentation from a perfectly reproducible training. When not fixing the whole training process, two different interventions leading to equivalent accuracy decrease could lead to equivalent churn. This for instance would be a direct consequence of a binomial modelisation of the model performance as a function of test set size and model average accuracy (and by the way which models fairly well test accuracy variation for the datasets-architectures in this paper). The lower the accuracy, the higher the variance.

> Table 2. We boldfaced the results in table 2 with the best mean performance, which we believe is a standard practice.

It is unfortunately common practice indeed, but it is a bad practice. Results that are so close that random fluctuations could explain the difference should not be considered as different.

---

> ### Author Response · Authors · 2020-11-17
> **Thanks for the comments**
>
>  We thank the reviewer and have updated the draft based on their comments and suggestions.
>
> 1. **Data order and augmentation in Ablation**: We agree with the reviewer that ablation experiments that ensure the same data order and augmentation across runs will give us a better idea about the cause of Churn. Indeed we have included such experiments in the appendix (Section E.1) of the submission, but on GPUs. By keeping the same data order and data augmentation across runs, we are able to preserve the accuracy and reduce churn (from 8.5% to 1.3%). Our experimental results agree with the reviewer’s experience in this regard. We faced certain technical difficulties in repeating similar experiments on TPUs, hence restricted them to the setting without data augmentation. We also emphasize that algorithms with lower accuracy do not necessarily have higher churn. For example, using a fully deterministic training method, one can achieve zero churn, irrespective of the accuracy value.
>
>  Given the experiments in the paper were on TPU, we felt it is appropriate to include the ablation studies on TPU in the main paper. We are in the process of replicating similar experiments on TPU and will include them in the final version. We note that this highlights the challenges in having fully reproducible predictions in training deep models and it is beneficial to develop techniques, independent of software/hardware, for reducing churn to address such concerns.
>
>
> 2. **Motivation**: There seems to be a confusion about the goal of the paper: we are interested in reducing the variance in individual predictions, not variance in accuracy across test sets. Our motivation is to ensure that predictions of the model do not change based on the randomness in training, such as initialization, mini batch ordering etc, even though the aggregate accuracy remains stable. Having predictions that are not sensitive to the random seed used in training is important for reproducibility, which is a very desirable property of machine learning models as established in existing works [2-4]. For example, a deterministically trained model can have lower test accuracy but will have zero churn as the model predictions do not change with different training runs. This property is independent of the test set size in this case.
>
> 3. **Hyperparameters**. We note that the learning rate, momentum,learning rate schedule, weight decay and the temperature for distillation parameters were chosen based on best performance of the baseline. For example, our baseline accuracy for Imagenet using Resnet50 matches the results from [1]. We **do not tune** these parameters for our experiments with the proposed methods. Given that we chose all the above key parameters based on best performance of the baseline, we believe this is a fair comparison. We experiment with alpha in range [0.05, 0.5] with 0.05 increments and beta in range [0.01, 0.08] with 0.01 increments. These details are given in Section A in appendix.
>
>  For reporting the results, we chose the value with the lowest churn with accuracy above the baseline. The performance of the proposed methods for different values of alpha and beta are presented in Figures 1 and 2 (comparing with existing approaches).
>
> 4. **Table 2**. We boldfaced the results in table 2 with the best mean performance, which we believe is a standard practice. We provide additional comparisons for varying alpha and beta in Figures 1 and 2.
>
> 5. **Co-distillation process**. In co-distillation we always pick model 1, assigned arbitrarily at the beginning of the training. We do not compare between the 2 models that are trained together.
>
> [1] https://pytorch.org/docs/stable/torchvision/models.html
> [2] Peter Henderson, Riashat Islam, Philip Bachman, Joelle Pineau, Doina Precup, and David Meger.Deep reinforcement learning that matters. In Thirty-Second AAAI Conference on Artificial  Intelligence, 2018.
> [3] Quentin Cormier, Mahdi Milani Fard, Kevin Canini, and Maya Gupta. Launch and iterate: Reducing prediction churn. In Advances in Neural Information Processing Systems 29, pp. 3179–3187. 2016.
> [4] Andrew Cotter, Heinrich Jiang, Maya Gupta, Serena Wang, Taman Narayan, Seungil You, and Karthik Sridharan. Optimization with non-differentiable constraints with applications to fairness, recall, churn, and other goals. Journal of Machine Learning Research, 20(172):1–59, 2019.

---

### Author Response · Authors · 2020-11-17
**Updates to the paper**

We thank all the reviewers for their thoughtful comments. Following the suggestions, we have included the following additional experiments in the paper.

1. **SVHN and iNaturalist** : We include experimental results on 2 additional datasets in Table 2. We notice again that the proposed methods are effective at reducing Churn while improving accuracy.
2. **Calibration** : We compute the Expected Calibration Error of the methods considered in the paper and include the results in Table 4 (Section 5), adding to our discussion on calibration. We note that while entropy regularizers, predictably, increase the calibration error, our combined approach with co-distillation results in calibration error competitive with the 2-ensemble distillation methods using only 2/3 of the training cost.
3. **Weight decay ablation** : All the models in the paper were trained with standard regularization techniques such as dropout and weight decay. We include ablation studies for models trained with and without weight decay regularization in Table 3, showing the effectiveness of the proposed approaches in both the settings.

---

### Decision · Program_Chairs · 2021-01-07
**Final Decision**

**Decision:**

Reject

**Comment:**

Fitting a neural net is a stochastic process, with many sources of stochasticity, including initialization, batch presentation, data augmentation, non-deterministic low-level operations and the non-associativity of rounding errors in multi-threads systems such as GPUs and TPUs. In this paper, the authors aim to alleviate this randomness by incorporating specific regularizers during learning or by using co-distillation.

As the reviewers pointed out, the paper is quite clearly written, but the motivation for this work is not clear. The example of system updates does not correspond to the current study that targets the internal variability of the learning process. Reproducibility is an important issue, but in a statistical context, why would it be relevant to assess reproducibility by the individual decisions made by a single estimate? The usual way of assessing learning algorithms is to look at (a summary of) the distribution of performance for a given learning problem characterized by a data distribution, not to look at individual decisions made by a particular estimate. Furthermore, this study ignores the randomness due to the selection of hyper-parameters. Why would the partial reproducibility studied here, for a fixed choice of hyper-parameters, be of particular interest? As is, either the work is ill-defined and incomplete, or it lacks a clear rationale, and I thus recommend rejection.

I would also like to point a reproducibility issue in the proposed experimental study. The exact meaning of the variability measures reported in the tables is not given, but I assume that it is the standard deviation of the different runs (for example, the 5 replicates in Table 1). These figures are not directly related to the variability of each setup, as they ignore the variability due to the random selections during the ablation study (for example, as I understand it, the last result of Table 1 was obtained for a single arbitrary initialization and a single arbitrary batch order).